# Paramagnetic relaxation enhancement NMR as a tool to probe guest binding and exchange in metallohosts

Anne Swartjes [1], Paul B. White [1 ✉], Jeroen P. J. Bruekers[1], Johannes A. A. W. Elemans [1 ✉] & Roeland J. M. Nolte [1 ✉]

Paramagnetic metallohost systems can bind guest molecules and find application as biomimetic catalysts. Due to the presence of the paramagnetic metal center, rigorous characterization of these systems by NMR spectroscopy can be very difficult. We report here that metallohost-guest systems can be studied by using the paramagnetic relaxation enhancement (PRE) effect. Manganese(III) porphyrin cage compounds are shown through their PRE to thread and bind viologen guests, including a polymeric one. The binding constants and dethreading activation parameters are lower than those of the metal-free porphyrin cage compounds, which is proposed to be a result of charge repulsion of the trivalent metal center and dicationic viologen guest. The threading rate of the manganese(III) porphyrin cage onto the polymer is more than 10 times faster than that of the non-metallated one, which is ascribed to initial binding of the cage to the polymer chain prior to threading, and to an entron effect.

[1] Institute for Molecules and Materials, Radboud University, Heyendaalseweg 135, 6525 AJ Nijmegen, The Netherlands. ✉email: p.white@science.ru.nl; j.elemans@science.ru.nl; r.nolte@science.ru.nl

NMR spectroscopy is a powerful tool in supramolecular chemistry, not only for characterizing host-guest systems and performing binding studies, but also for investigating the kinetics of host–guest exchange at equilibrium[1–5]. Diamagnetic systems are the simplest to study since they display sharp lines, allowing for a multitude of 1D and multidimensional techniques to be applied. Paramagnetic systems pose complications for studying host–guest interactions since chemical shift and nuclear relaxation can be greatly influenced by the presence of unpaired electrons. However, even these challenging properties can provide opportunities for answering chemical questions and be utilized to gain insight into exchange and binding processes. Here, we show that metallohost–guest binding and exchange in a paramagnetic manganese(III) porphyrin macrocycle can be studied by NMR spectroscopy by exploiting the paramagnetic properties of the metal center, i.e., by using the paramagnetic relaxation enhancement (PRE) effect. So far, PRE has been utilized by various NMR methods studying paramagnetic systems such as characterization by means of hyperfine shifts[6–8], measurement of protein–ligand kinetics via dissolution dynamic nuclear hyperpolarization (DNP)[9], studying Xe host-guest complexes via paramagnetic hyper-CEST[10, 11], paramagnetic GEST (host–guest variant of CEST) combining MRI and NMR[12], and diffusion NMR (DOSY)[13, 14]. Furthermore, a toolbox using standard NMR techniques has been reported in the characterization of paramagnetic compounds[15]. In biochemistry, PRE has mainly been applied for the study of enzyme mechanisms[16–18], protein structure characterization[19–22], drug discovery[23–27], and protein dynamics and exchange reactions[28–33]. To the best of our knowledge PRE NMR has not been used to study association and dissociation phenomena in organic and supramolecular chemistry.

The studies presented here are part of an overarching project aimed at the development of manganese(III) porphyrin macrocyclic catalysts that can thread onto a polybutadiene chain, and while moving along it, convert the polymer double bonds into epoxides[34, 35]. In previous work, we studied the mechanism of exchange of small N,N′-dialkyl-4,4′-bipyridinium (viologen) guests as well as polymers containing a viologen moiety in diamagnetic (metal-free or Zn) porphyrin macrocycles through UV–Vis, fluorescence, and EXSY NMR spectroscopy[36]. Unfortunately, spectroscopic studies on the actual catalytic system, i.e., the Mn(III)-porphyrin macrocycle, have not been possible, and as a result the effect of the Mn center in the porphyrin on exchange and binding has not yet been determined. This is due to the lack of fluorescent properties of Mn porphyrins, inconclusive UV–Vis behavior, and short $T_2$s, which cause unmanageably broad peaks in the $^1$H NMR spectra, making EXSY studies impossible to perform. Hence, we decided to study the dynamics of the actual host–guest system by measuring the PRE on an excess of free

viologen guests, including both low molecular weight and polymeric ones (V1, V2 and VP, see Fig. 1), which are exchanging at equilibrium with the bound host Mn(III)-porphyrin macrocycle Mn1. In this work, we provide a clear-cut NMR spectroscopic method to study exchange in these paramagnetic host–guest systems and to determine the threading and binding parameters involved. We show that manganese(III) porphyrin cage compound Mn1 can thread and bind viologen guests, including a polymeric one. The binding constants and de-threading activation parameters unexpectedly are lower than those of the metal-free porphyrin cage compound $H_2$1. Furthermore, the threading rate of Mn1 onto the polymer is much faster than that of $H_2$1, which is ascribed to initial binding of the cage to the polymer chain prior to threading, and to an entron effect.

## Results

In the presence of a paramagnetic center, an increase in the longitudinal ($R_{1,obs}$ (s$^{-1}$)) and transverse ($R_{2,obs}$ (s$^{-1}$)) relaxation rate of a given nucleus can be observed. This observed relaxation rate is defined as the sum of the diamagnetic ($R_0$) and paramagnetic ($R_p$) relaxation rates (Eq. 1). $R_{1,obs}$ and $R_{2,obs}$ can be determined by measuring the $T_1$ (s) and $T_2$ (s) values of a specific signal in a sample containing both the paramagnetic host and diamagnetic guest compound in solution. The terms $R_{1,0}$ and $R_{2,0}$ can be obtained by measuring the $T_1$ and $T_2$ values of this signal in a sample containing only the diamagnetic guest compound in solution. Exchange parameters can be calculated from $R_p$, which, however, is dependent on the regime of exchange.

$$R_{obs} = R_0 + R_p \qquad (1)$$

There are three regimes of exchange (fast, intermediate, and slow) with respect to nuclear relaxation that provide information about the interaction of a guest with a paramagnetic center.

Here, we take advantage of PRE in two ways: (1) fast exchange of solvent inside an empty host to determine the association constant of viologen guest binding, and (2) slow, reversible dissociation of viologen guests to determine dissociation rate constants.

**Measuring the binding strength of paramagnetic host-guest systems.** The $^1$H NMR spectrum of the diamagnetic metal-free porphyrin macrocycle $H_2$1 gives well-defined and isolated sets of signals (Fig. 2, top). However, when a Mn(III)-Cl center is inserted into the porphyrin (Mn1), all host signals broaden due to the paramagnetic effect of the metal center on the cage compound (Fig. 2, bottom). The sharpest of the signals, near 7.0 ppm, correspond to the phenyl groups at the bottom of the cage, which are the farthest away from the Mn center and thus

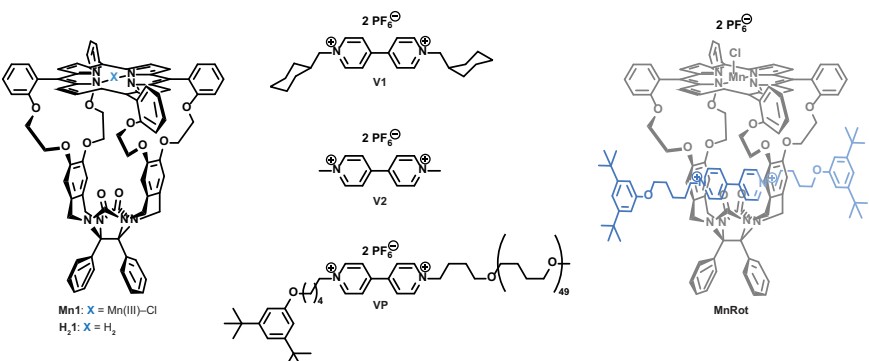

**Fig. 1 Molecular structures of the studied compounds.** Mn(III)-porphyrin cage compound **Mn1** (porphyrin center atom(s) indicated in blue), bis-(methyl cyclohexyl) viologen **V1**, methyl viologen **V2**, polymer-substituted viologen **VP**, and the rotaxane of a blocked viologen guest (blue) with **Mn1** (**MnRot**).

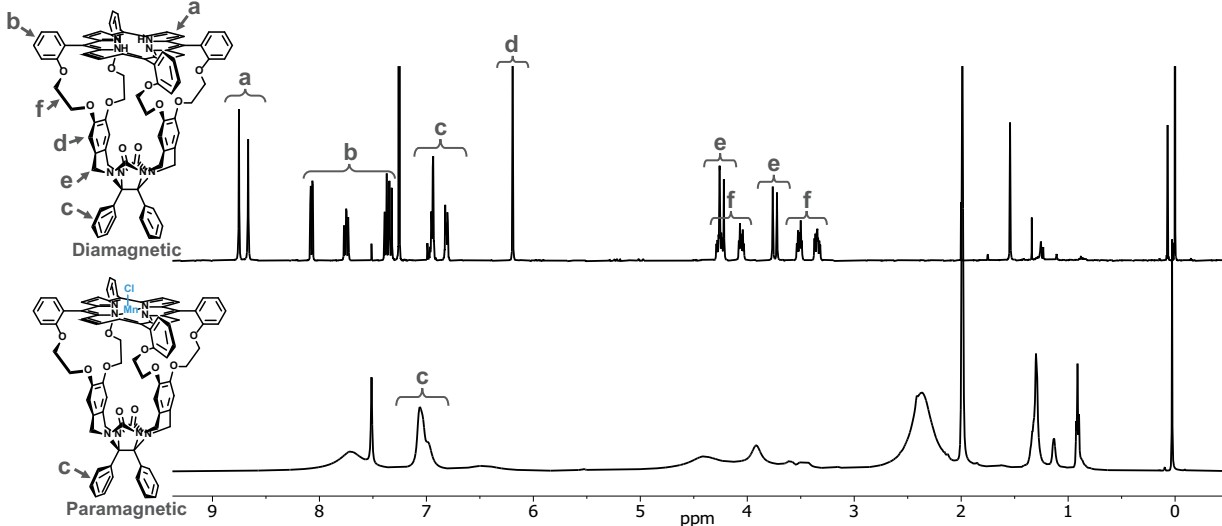

**Fig. 2 Influence of the presence of a paramagnetic Mn center on the $^1H$ NMR spectrum of a host compound.** Top: $^1H$ NMR spectrum of metal-free porphyrin cage compound $H_2 1$ (500 MHz, chloroform-$d$: acetonitrile-$d_3$, 1:1 (v/v), 25 °C), bottom: $^1H$ NMR spectrum of **Mn1** (500 MHz, chloroform-$d$: acetonitrile-$d_3$, 1:1 (v/v), 25 °C) (metal center and counter ion in blue).

| Table 1 Binding constants of host–guest complexes. | | | |
|---|---|---|---|
| **Host** | **Guest** | **Log($K_a$)** | **$\Delta G$ (kcal mol$^{-1}$)$^e$** |
| $H_2 1$ | V1 | $7.29 \pm 0.26^a$ | $-9.95 \pm 0.16$ |
| Mn1 | V1 | $3.68 \pm 0.16^b$ | $-5.02 \pm 0.10$ |
| $H_2 1$ | V2 | $5.78 \pm 0.30^{a,c}$ | $-7.88 \pm 0.18$ |
| Mn1 | V2 | $4.20 \pm 0.17^b$ | $-5.72 \pm 0.11$ |
| $H_2 1$ | $VP_{var}$ | $\sim 7.4^{a,d}$ | $-10$ to $-11$ |
| Mn1 | VP | $3.23 \pm 0.18^b$ | $-4.40 \pm 0.10$ |

$^a$Binding constants were determined by UV–Vis titrations in CHCl$_3$/CH$_3$CN 1:1 (v/v).
$^b$Binding constants were determined by PRE NMR titrations in CHCl$_3$/CDCl$_3$/CD$_3$CN 1:4:5 (v/v/v).
$^c$Value taken from ref. 33.
$^d$Value taken from refs. 40, 41 at 23 °C for a similar polymeric structures as **VP**, which, however, had variable poly-THF chain lengths (34–88 monomer units).
$^e$Calculated for $T = 25$ °C.

experience the least dipolar relaxation enhancement due to the $r^{-6}$ distance component. In order to accurately calculate the exchange parameters of the host–guest systems, the population of bound host should be determined for each measured system, which can be calculated through the respective binding constants. Due to the difficulty of measuring binding through routinely-used spectroscopic techniques[36–40], we resorted to biochemical NMR spectroscopy for inspiration, as our hosts mimic to some extent properties of metalloproteins and metalloenzymes. Claridge et al. have taken advantage of the change in $R_{1,obs}$ of the bulk solvent water when it has access to a paramagnetic metal center buried inside a protein to determine the binding constant of a second, more competitive ligand to this metal center[41]. As soon as the competitive ligand binds to the metal center completely, the water will only be affected by the weaker outer-sphere effects, resulting in a smaller $R_{1,obs}$.

We decided to perform PRE NMR studies on our host-guest complexes to measure binding constants, but instead of measuring the longitudinal relaxation rate of water, we used chloroform as a reporter solvent and acetonitrile as co-solvent to improve guest solubility. Chloroform has been observed in X-ray structures of $H_2 1$ to reside inside the cavity of the host and thus is an ideal reporter solvent[42]. Acetonitrile was also tested as a reporter solvent, but upon binding it displayed an overall smaller change in the relaxation rate, reducing its reliability. A binding

curve was constructed by measuring the relaxation rate of bulk chloroform while varying the guest concentration and keeping the **Mn1** concentration constant. From these PRE binding titrations, the binding constants of the **Mn1/V1**, **Mn1/V2**, and **Mn1/VP** complexes could be determined (Table 1, for titration curves see Supplementary Fig. 21).

As shown in Table 1, the log($K_a$) of the paramagnetic host–guest complex **Mn1/V1** ($3.68 \pm 0.16$) is significantly lower, i.e., by three orders of magnitude, than that of the diamagnetic host–guest complex $H_2 1$/**V1** ($7.29 \pm 0.26$). This indicates that $H_2 1$/**V1** forms a much more stable complex than **Mn1/V1**. This decrease in binding affinity after manganese insertion may be explained by repulsion between the positively charged dicationic viologen guest molecule, and the trivalent metal center which will destabilize the binding of **V1** inside the cavity. In this connection it is worth mentioning that the electronic structures of manganese(III) porphyrins, e.g., Mn(III)TPP Cl, have been widely studied in the literature. A partial charge build-up on the metal center was found to occur as a result of the relatively labile Mn−Cl bond[43–45].

As expected, the binding constant for **Mn1/V2** (log($K_a$) = $4.20 \pm 0.17$) was lower than that of the diamagnetic combination $H_2 1$/**V2** (log($K_a$) = $5.78 \pm 0.30$)[39]. However, **V2** binds more strongly inside **Mn1** than **V1**, whereas the opposite was true for the diamagnetic host $H_2 1$. This could be caused by the decrease in sterics when the bulkier methyl cyclohexyl groups on either side of the viologen moiety in **V1** are replaced by methyl groups, allowing the resulting guest **V2** to be positioned more favorably inside the cavity of **Mn1**. This may result in a slight reduction of repulsive interactions between the metal-center and **V2**, in contrast to **Mn1/V1** where there is less freedom of movement.

Lastly, the binding constant for **Mn1/VP** (log($K_a$) = $3.23 \pm 0.18$) was found to be similar in magnitude to **Mn1/V1** (log($K_a$) = $3.68 \pm 0.16$). This is consistent with our previous research where binding constants were determined for $H_2 1$ and poly-THF substituted viologen guests with variable numbers of monomer units (34–88)[46, 47]. The obtained binding constants (log($K_a$) ~7.4) were very similar to that of the small, symmetric viologen guest **V1** with $H_2 1$ (log($K_a$) = $7.29 \pm 0.26$), indicating that similar (de)stabilizing effects are present. A range of differently metalated porphyrin cages is currently under investigation. A step-by-step how-to guide for determining

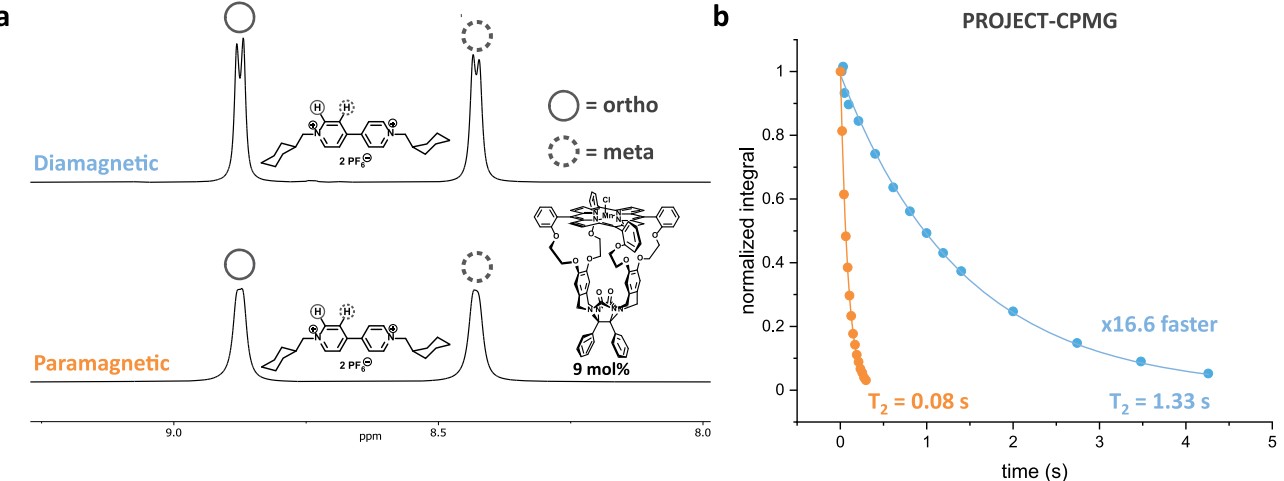

**Fig. 3 Influence of the presence of Mn1 on $T_2$ values of free guest proton signals. a** Top: [1]H NMR spectrum of **V1** (500 MHz, chloroform-*d*: acetonitrile-*d*₃, 1:1 (v/v), 20 °C), bottom: [1]H NMR spectrum of **V1** in the presence of 9 mol% **Mn1** (500 MHz, chloroform-*d*: acetonitrile-*d*₃, 1:1 (v/v), 20 °C). **b** Mono-exponential first order plots of PROJECT-CPMG measurements and the determined $T_2$ values for the relaxation of the unbound meta-protons of **V1** and of **V1** in the presence of 9 mol% **Mn1** (20 °C), showing an enhancement factor of 16.6×. The color-coding blue and orange refer to the experiments without (diamagnetic) and with (paramagnetic) **Mn1** present, respectively.

association constants and dissociation parameters is included in the Supplementary information.

**Exchange of viologen guests in Mn1**. It is evident from Fig. 2 that selective [1]H-NMR experiments, such as 1D EXSY, on host-guest complexes involving **Mn1** are not possible due to the excessive broadening of the relevant host resonances. Upon adding ten equivalents of a viologen guest **V1** to **Mn1**, we can observe a similar paramagnetic-induced broadening of the ortho (8.85 ppm) and meta (8.45 ppm) free guest proton resonances compared to the resonances in the diamagnetic host-free spectrum (Fig. 3a). Notably, the bound **V1** signal is absent, most likely due to a strong $R_2$ PRE. Fortunately, the limited broadness still allows relaxation experiments to be performed with relative ease on the free guest. Preliminary $T_2$ measurements revealed a > 16× faster relaxation for **V1** (5 mM) in the presence of **Mn1** (0.5 mM) compared to the host-free condition (Fig. 3b). The magnitude of such a large enhancement suggests the occurrence of exchange-based PRE rather than a relaxation enhancement purely caused by the presence of a paramagnetic species in absence of exchange (outer-sphere effects), warranting further investigations[16].

As described in Eq. 1, $R_{obs}$ can be divided into a diamagnetic ($R_0$) and a paramagnetic contribution ($R_p$). $R_p$ can be further defined as the sum of inner-sphere (close, long-lived binding interactions, $\frac{f}{\tau_M + T_M}$) and outer-sphere interactions (distant, non-binding interactions, $R_{os}$) (see Fig. 4a). Here, "f" is the the mole-fraction of the ligand bound to the host that takes into account the binding affinity of the guest molecule to the host and the number of binding sites, $\tau_M$ (s) is the mean-residence time of the guest bound to the host, and $T_M$ (s) is the relaxation time constant the guest experiences while bound. By taking the inverse of $\tau_M$, the observed dissociation rate constants can be determined and thereby their corresponding energy barriers. When the system falls into a slow-to-intermediate exchange regime with respect to relaxation, $\tau_M$ can dominate $R_p$ under certain conditions and therefore the $T_M$ term can be neglected, simplifying the equation to Eq. 2.

$$R_p = \frac{f}{\tau_M} + R_{os} \qquad (2)$$

In order to determine if $R_p$ is dominated by $\tau_M$, and therefore suitable for measuring the rate of dissociation of the guest from the host, three requirements have to be met: 1) $R_p$ should increase at higher temperatures ($\frac{\partial R_p}{\partial T} > 0$) since $\tau_M$ would decrease with faster exchange, 2) $R_p$ is independent of the field strength ($\frac{\partial R_p}{\partial \omega} = 0$) since $\tau_M$ is field-independent, and 3) $R_{1,p}$ should approximately be equal to $R_{2,p}$ ($\frac{R_{2,p}}{R_{1,p}} \approx 1$)[16]. A more in-depth explanation and justification of the requirements and the simplification of the aforementioned equations can be found in the Supplementary Information.

To determine whether the above requirements are met, the paramagnetic relaxation rates $R_{1,p}$ and $R_{2,p}$ of the **Mn1/V1** system were first measured at both 300 and 500 MHz and from 30 to 50 °C. As shown in Fig. 4b–d, all requirements were suitably met within the above temperature range, indicating that $\tau_M$ was indeed dominating $R_p$. Subsequently, the temperature range was extended to the range −10 to 70 °C in order to be able to assess the kinetics over the largest possible temperature span for an Eyring analysis. When the measurements were performed at lower temperatures, it was observed that the "$\frac{R_{2,p}}{R_{1,p}} \approx 1$" condition began to deteriorate as $R_{2,p}/R_{1,p}$ began to approach 1.5 at −10 °C. This suggested that outer-sphere contributions were playing a larger role in $R_p$ as $\tau_M$ became longer at lower temperatures. To correct for these outer-sphere effects, a manganese rotaxane (**MnRot**, Fig. 1), where the binding cavity is irreversibly blocked, was synthesized (see Supplementary Information), and $R_{1,obs}$ and $R_{2,obs}$ were measured under identical conditions for **MnRot/V1** over the complete temperature range. After correcting for the outer-sphere interactions, the "$R_{1,p} \approx R_{2,p}$" condition proved to be satisfactory held over the complete temperature range (Fig. 4d).

By measuring $R_{obs}$, $R_0$, and $R_{os}$ the observed dissociation rate constant ($k_{d,obs}$) for **V1** could be calculated using Eqs. 1 and 2 (Tables S8 and S9). To calculate the actual dissociation rate constant ($k_d$), the observed dissociation rate constant has to be corrected for the host and guest dependence on the residence time of the ligand near or on the metal ($\tau_M$). If both the host and guest concentration have no influence on $\tau_M$, $k_{d,obs}$ will be equal to $k_d$. However, if there is an additional host or guest dependence on the residence time, the $k_{d,obs}$ will have to be corrected for the order of the dependence. We determined the host and guest dependence on the exchange by

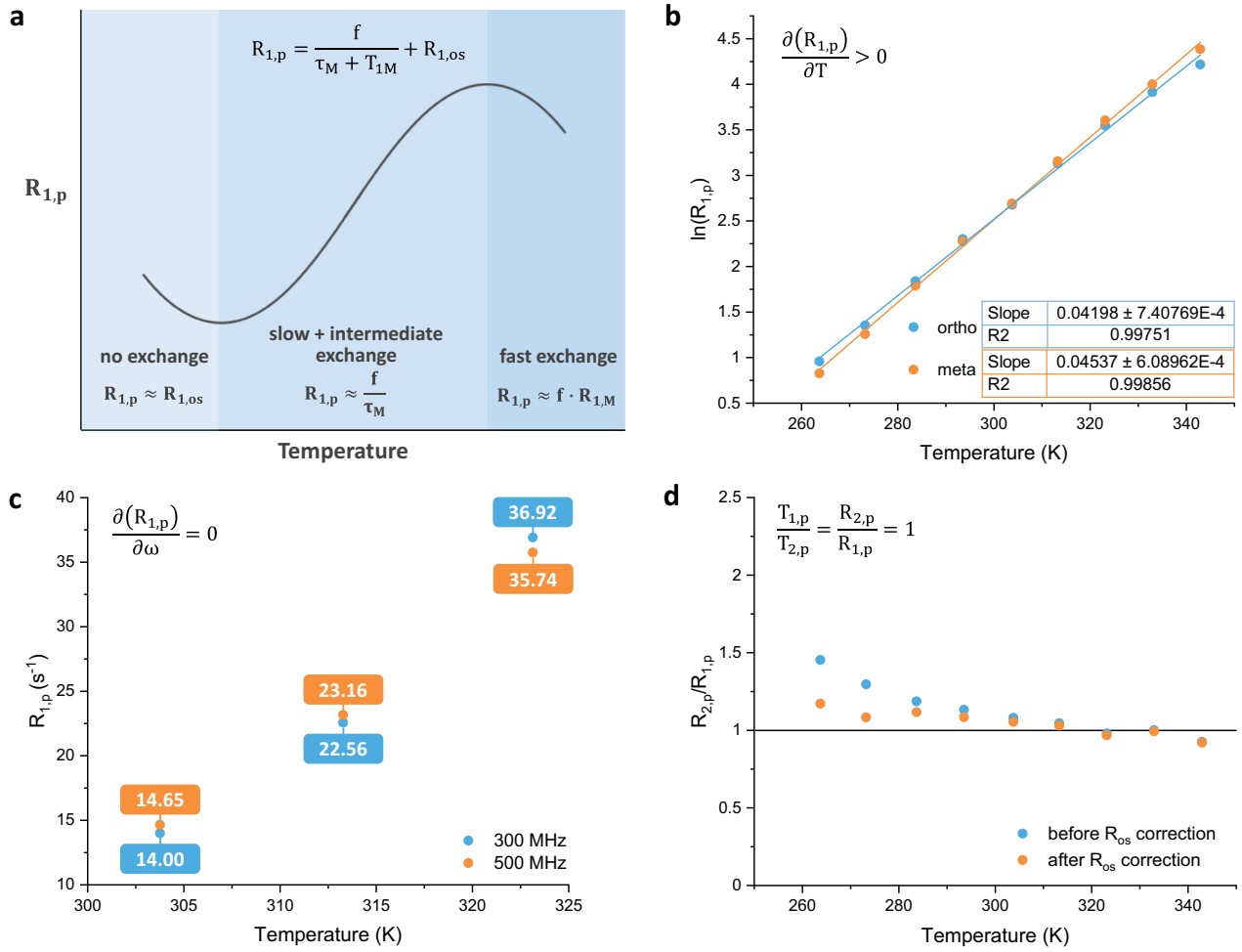

**Fig. 4 Determination of relaxation rates of the Mn1/V1 system. a** Schematic indication of the different dominating factors governing exchange at different temperatures (low $T$ is left, high $T$ is right). At low temperatures where there is no exchange, the outer-sphere interactions are dominating; at higher temperatures the slow and intermediate exchange regime holds, and the exchange is dominating; and in the fast exchange regime the inner-sphere interactions dominate. The various regimes are indicated in three shades of blue. **b** Graph of $\ln(R_{1,p})$ against the temperature, displaying a positive slope with increasing temperature. The data points for the ortho protons are shown in blue and those for the meta protons in orange. **c** $R_{1,p}$ values plotted against temperature comparing 300 MHz (blue dots and blue numbers) and 500 MHz (orange dots and orange numbers) experiments. **d** Ratio of $R_{2,p}$ and $R_{1,p}$ plotted against temperature before (blue dots) and after (orange dots) $R_{1,os}$ and $R_{2,os}$ correction. Solvent chloroform-$d$: acetonitrile-$d_3$, 1:1 (v/v).

measuring the residence time of the guest in the **V1/Mn1** complex at different **V1** and **Mn1** concentrations (see Supplementary Table 7). From these experiments an exchange-promoting behavior was found when the guest concentration was increased. However, the exchange process was characterized by a non-integer order (~0.3). When the measurements were repeated for **Mn1** and a guest similar to **V1** (Supplementary Fig. 2b), again a non-integer order (~0.4) was obtained. To determine whether this guest dependence resulted from a guest or a host effect, EXSY experiments were performed on the diamagnetic **H₂1/V1** complex (Supplementary Figs. 23 and 24)[36]. In this case, a zero-order guest dependence was found (Supplementary Fig. 2c), indicating that the presence of the manganese metal inside the porphyrin ring significantly alters the mechanism for guest dissociation. The observed non-integer dependence can have different origins: the possible additional binding of a guest molecule on the outside of the porphyrin cage could promote dissociation, the counter ions of the viologen salt (hexafluorophosphates) could play a role by exchanging with Cl⁻ ions, which may influence the ability for the guest to dissociate, and, finally, a combination of associative and dissociative mechanisms may occur. These mechanistic explanations are interesting and require further studies, which are underway. Our primary interest, however, was the determination of the activation

**Table 2 Activation enthalpies, entropies and free energies of dissociation for different host-guest complexes.**

| Host | Guest | $\Delta H^{\ddagger}$ (kcal mol⁻¹) | $\Delta S^{\ddagger}$ (cal K⁻¹ mol⁻¹) | $\Delta G^{\ddagger}$ (kcal mol⁻¹)[a] |
|------|-------|------|------|------|
| $H_2$1 | V1 | 10.50 ± 0.29 | −25.10 ± 1.06 | 17.99 ± 0.43 |
| Mn1 | V1 | 8.35 ± 0.17 | −17.57 ± 0.34 | 13.59 ± 0.17 |
| Mn1 | V2 | 7.99 ± 1.08 | −5.60 ± 0.35 | 9.65 ± 1.08 |
| Mn1 | VP | 3.17 ± 0.33 | −36.89 ± 3.29 | 13.92 ± 0.40 |

[a]Calculated for $T = 25\,°C$.

parameters for guest dissociation from the host–guest complex. After correcting $k_{d,obs}$ for the guest dependence, the activation energies ($\Delta H^{\ddagger}$, $\Delta S^{\ddagger}$, and $\Delta G^{\ddagger}$) were extracted from the Eyring plots (Table 2 and Supplementary Fig. 20).

In order to assess the influence of Mn(III) Cl insertion into the porphyrin ring we may compare the activation energy barriers for dissociation of the **Mn1/V1** system with those of the diamagnetic system **H₂1/V1**. Interestingly, the $\Delta G^{\ddagger}$ of the **Mn1/V1** complex is substantially lower (13.59 ± 0.17 kcal mol⁻¹) than that of the diamagnetic host–guest complex **H₂1/V1** (17.99 ± 0.43 kcal mol⁻¹). The difference of ~4.5 kcal

**Table 3** $k_{on}$ **values and threading parameters for Mn1/VP, and reference parameters for H$_2$1 with comparable poly-THF substituted viologen guests.**

| Host | #THF-units in VP | $k_{on}$ (M$^{-1}$ s$^{-1}$) | $\Delta H^{\ddagger}$ (kcal mol$^{-1}$) | $\Delta S^{\ddagger}$ (cal K$^{-1}$ mol$^{-1}$) | $\Delta G^{\ddagger}$ (kcal mol$^{-1}$) |
|---|---|---|---|---|---|
| H$_2$1 | 34 | $3.6 \times 10^4 \pm 20\%$[a] | $4.78 \pm 15\%$[a] | $-21.03 \pm 10\%$[a] | $11.23 \pm 5\%$[a,c] |
| H$_2$1 | 60 | $1.9 \times 10^4 \pm 20\%$[a] | $4.78 \pm 15\%$[a] | $-22.47 \pm 10\%$[a] | $11.61 \pm 5\%$[a,c] |
| Mn1 | 49 | $(4.66 \pm 1.04) \times 10^5$ [b,c] | $3.17 \pm 0.64$[d] | $-22.09 \pm 3.94$[d] | $9.76 \pm 1.35$[c,e] |

[a]Values taken from refs. [40, 41] ($k_{on}$ values measured at 23 °C).
[b]The $k_{on}$ values were calculated from $K_a = \frac{k_{on}}{k_d}$; $k_d$ values can be found in Tables S10 and S11.
[c]Values for $T$ = 25 °C.
[d]Determined from the Eyring plot with calculated $k_{on}$ values over the temperature range 20–70 °C, assuming that the $K_a$ of **Mn1/V1** remains constant over this temperature range.
[e]Calculated from $\Delta G^{\ddagger} = -RT\ln\left(\frac{k_{on}h}{k_B T}\right)$ in which $R$, $h$, $k_B$ are the ideal gas constant, the Planck constant, and the Boltzmann constant, respectively.

mol$^{-1}$ in $\Delta\Delta G^{\ddagger}$ mirrors the ~5 kcal mol$^{-1}$ difference in $\Delta\Delta G$ (Table 1), suggesting that ground-state effects play a significant role in lowering the activation energies for exchange when a Mn(III) ion is inserted into the porphyrin of the cage compound.

The PRE experiments were repeated with **Mn1** and viologen polymer **VP**. This guest molecule has been studied previously with **H$_2$1**, where it was confirmed that threading and exchange do occur[36]. We previously had expected and assumed that polymers would thread through **Mn1**[47, 48], but PRE NMR studies now offer the possibility to definitively prove whether this hypothesis holds true. For the **Mn1/VP** system $R_{obs}$ should be compared to $R_{os}$. Here, $R_{obs}$ is measured by a sample containing **Mn1** and **VP**, and for $R_{os}$ a sample containing **MnRot** and **VP**. If $R_{obs} \approx R_{os}$ then the inner-sphere contribution is near-zero and there will be (almost) no exchange and, consequently, possibly no threading. If the observed relaxation rate is much larger than the outer-sphere contribution ($R_{obs} \gg R_{os}$), then this supports that exchange and hence threading occurs. We performed these PRE NMR experiments and found that the observed relaxation rates ($R_{1,obs}$ = 11.94 s$^{-1}$, $R_{2,obs}$ = 10.02 s$^{-1}$) indeed are significantly larger than the outer-sphere contributions ($R_{1,os}$ = 1.03 s$^{-1}$, $R_{2,os}$ = 1.38 s$^{-1}$), i.e., by approximately a factor of nine, which is in a similar order of magnitude as found for **Mn1/V1**. These experiments, therefore, demonstrate that **Mn1** indeed threads **VP**.

From the Eyring plot (Fig. S20c, d) the activation energy barriers for dissociation could be determined (Table 2), and similar energy barriers were observed for **Mn1/VP** (13.92 ± 0.40 kcal mol$^{-1}$) compared to **Mn1/V1** (13.59 ± 0.17 kcal mol$^{-1}$). The notion that the destabilized binding of **V1** in the cavity of **Mn1** results in facile dissociation and therefore faster exchange is further confirmed by the measured binding constants of **Mn1/VP** ($K_a$ = 1692 ± 295 M$^{-1}$) and **Mn1/V1** ($K_a$ = 4796 ± 769 M$^{-1}$), which are in the same order of magnitude. Furthermore, since chloroform exposure to the cavity was the observable feature in the binding studies, these experiments directly point to complete dissociation of the polymer as opposed to a slipping process in which the viologen moiety slips in and out of the cavity without full dissociation of the polymer-chain.

From the measured energy barriers for exchange ($\Delta H^{\ddagger}$ = 3.17 ± 0.33 kcal mol$^{-1}$, $\Delta S^{\ddagger}$ = $-36.89 \pm 3.29$ cal K$^{-1}$ mol$^{-1}$) it is evident that for the **Mn1/VP** system the dissociation is almost entirely governed by entropy. Possible reasons for the high entropy contribution to the dissociation of the polymer could be linked to the notion that a significant extent of order should be achieved for complete dissociation of a molecule with this length. It could also be influenced by the resolvation of the polymer chain after dissociation or possible wrapping of the polymer-chain around the porphyrin host[37]. The significantly lower activation enthalpy ($\Delta H^{\ddagger}$ = 3.17 ± 0.33 kcal mol$^{-1}$) found for **Mn1/VP** compared to **Mn1/V1** ($\Delta H^{\ddagger}$ = 8.35 ± 0.17 kcal mol$^{-1}$) may be attributed to the possibility that the viologen-moiety of **VP** is threaded, but not deeply bound inside the cavity of **Mn1**, as opposed to **Mn1/V1**, resulting in a lower dissociation activation enthalpy for **Mn1/VP**. Another possibility is the difference in sterics between **V1** and **VP**. The observed higher $\Delta H^{\ddagger}$ of the former compound may be a result of the presence of the bulkier methyl cyclohexyl substituents on either side of its viologen moiety compared to the unbranched carbon chains attached to that of **VP**, possibly complicating dissociation. To test the latter theory and to see if the activation enthalpy would be lower, the PRE NMR measurements were repeated with methyl viologen **V2** (see Supplementary Information p 35–40). Here, similar $\Delta H^{\ddagger}$ values for the dissociation of **Mn1/V2** (7.99 ± 1.08 kcal mol$^{-1}$) compared to that of **Mn1/V1** (8.35 ± 0.17 kcal mol$^{-1}$) were found, rejecting the hypothesis that less bulky substituents on the viologen guest result in lower activation enthalpies. Hence, we attribute the appreciable difference in dissociation $\Delta H^{\ddagger}$ values measured for **Mn1/V1** and **Mn1/VP** to an increase in statistical host positions on the polymer chain, as opposed to **Mn1/V1** where the viologen moiety is deeply bound inside the cavity of its host.

Lastly, from the now known $k_d$ values of **Mn1/VP** (Supplementary Tables 10 and 11) and the previously determined binding constant, threading rate constants ($k_{on}$) can be calculated from $K_a = \frac{k_{on}}{k_d}$ and be used to construct an Eyring plot from 20 to 70 °C, from which activation energies can be extracted. Here, it is assumed that the binding constant of **Mn1/VP** remains relatively insensitive to the temperature (see Supplementary Fig. 22). The calculated threading parameters for **Mn1/VP** are listed in Table 3, together with threading parameters from previous studies in which the threading of **H$_2$1** onto variable-length poly-THF substituted viologens was studied. It is apparent from Table 3 that **Mn1/VP** surprisingly threads faster ($k_{on}$ = (4.66 ± 1.04) × 10$^5$ M$^{-1}$ s$^{-1}$) than combinations of **VP** with the metal-free host ($k_{on}$ = (1.9–3.6) × 10$^4$ M$^{-1}$ s$^{-1}$)[46]. Analysis of the Eyring parameters reveals that the difference between the threading of **VP** through **Mn1** and through **H$_2$1** is solely enthalpic in nature, as the entropy values are very similar. We tentatively ascribe this to a pre-binding effect, which has been proposed in earlier work[47, 49]. It involves a preassociation of the porphyrin cage with the polymer and diffusion along its chain, allowing the polymer end to loop back into the cavity of the host (effective molarity effect), aiding the threading process. We suggest that this process is more effective for **Mn1/VP** than for **H$_2$1/VP** because of the positive charge residing on the **Mn1** center, interacting stronger with the oxygen atoms in the polymer chain. Furthermore, the interaction of the polymer end with the cavity of the cage, which initiates the threading process (entron effect)[47], will be stronger for **Mn1** than for **H$_2$1**. The above mechanism explains why the $\Delta S^{\ddagger}$ values for **Mn1/VP** and **H$_2$1/VP** are similar, but the $\Delta H^{\ddagger}$ value is lower for **Mn1**.

## Discussion

We have shown that paramagnetic relaxation enhancement experiments can be utilized to investigate chemical exchange binding strength in the field of metallohost–guest chemistry for systems displaying both slow to medium exchange rates (utilized to measure dissociation) and fast exchange rates (utilized to

perform association titrations), even if the resonances corresponding to the bound species are not directly visible. Binding titration studies revealed that a paramagnetic host–guest complex, involving a manganese(III) metal in the porphyrin ring of the cage compound, displays significantly lower binding affinities for viologen guests compared to those of the non-metallated porphyrin cage with the same guest compounds. Furthermore, **Mn1/V1** displays lower energy barriers for guest dissociation compared to **H$_2$1/V1**. The combined results of lower binding constants and lower dissociation barriers for **Mn1/V1** compared to **H$_2$1/V1** indicate that for the paramagnetic system ground-state effects play an important factor in the decrease of the activation barriers for dissociation. These destabilizing effects may be caused by repulsive interactions between the positively charged metal center and the dicationic viologen guest. Furthermore, decreasing the bulkiness on the substituents of the guest compound (compare **Mn1/V2** versus **Mn1/V1**) resulted in slightly stronger binding and faster dissociation, which likely is a result of the increased freedom of movement of the guest inside the cavity of the host in combination with decreased steric interactions.

The PRE studies could also be performed on the host–guest complex **Mn1/VP**, which involves a polymer chain to which a viologen moiety is attached. The much larger value observed for the relaxation rates ($R_{1,obs} = 11.94$ s$^{-1}$, $R_{2,obs} = 10.02$ s$^{-1}$) compared to the outer-sphere contributions ($R_{1,os} = 1.03$ s$^{-1}$, $R_{2,os} = 1.38$ s$^{-1}$) presents irrefutable evidence that **Mn1** is capable of threading **VP**, confirming a long-standing hypothesis. Surprisingly, the threading rate of **Mn1** onto **VP** was found to be considerably faster than that of the non-metallated porphyrin cage **H$_2$1** onto **VP**. We attribute this to a pre-association of the positively charged **Mn1** complex to the outside of the oxygen-containing poly-THF chain, allowing the polymer end to loop back and find the cavity of **Mn1** more easily (effective molarity effect). The methodology presented in this paper provides a blueprint for how binding and dynamics in other paramagnetic systems, especially catalytic ones, which are extensively studied in the field of organic chemistry, could be thoroughly investigated.

## Methods
Longitudinal relaxation rates were measured through inverse recovery experiments and transverse relaxation rates were measured via by utilizing the CPMG[50, 51] (Carr-Purcell-Meiboom-Gill) pulse sequence and the PROJECT-CPMG[52] (Periodic Refocusing of J Evolution by Coherence Transfer CPMG) sequence on a Bruker 500 MHz Avance III spectrometer equipped with a Prodigy BB cryoprobe or a Bruker 300 MHz Avance III HD nanobay spectrometer equipped with a BBFO probe. For all VT experiments, the temperature was calibrated using pure ethylene glycol for temperatures of 20 °C and higher. Methanol was used to calibrate the temperatures lower than 20 °C.

Fluorescence quenching titrations were performed on a JASCO FP-8300ST Spectrofluorometer in a quartz cuvette with 1 cm path length. Samples for titrations consisted of anhydrous chloroform and anhydrous acetonitrile (1:1, v/v). Anhydrous chloroform (>99%) and acetonitrile were obtained from Sigma-Aldrich. Before use, the chloroform was filtered over dry K$_2$CO$_3$ and acetonitrile was distilled and dried over CaCl$_2$. For all titration studies, which were carried out at 22 °C, the solutions were irradiated with light of 419 nm.

## Data availability
The authors declare that the data supporting the findings of this study are included in the paper and its Supplementary Information files. Any further relevant data are available from the corresponding authors on request.

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

## Acknowledgements

This work was funded by the European Research Council (ERC Advanced Grant No. 74092 to R.J.M.N.) and by the Dutch Ministry of Education, Culture, and Science (Gravitation program 024.001.035).

## Author contributions

R.J.M.N. conceived the project. A.S., P.B.W. and J.A.A.W.E. designed and performed the NMR experiments. J.P.J.B. performed titration experiments. All authors discussed the results and contributed to drafts of the manuscript.

## Competing interests

The authors declare no competing interests.
