## [Peer Review File · Nature Communications]

Paramagnetic relaxation enhancement NMR as a tool to probe guest binding and exchange in metallohostsREVIEWER COMMENTS

Reviewer #1 (Remarks to the Author):

What are the noteworthy results?

This manuscript describes the application of paramagnetic relaxation enhancements to probe the host-guest interactions between manganese(III)-porphyrin-based host and viologen-based guests. It is demonstrated that PRE data can be used to investigate chemical exchange and, in particular, to prove that the host can thread the guest, which is the most relevant finding in this paper.

Will the work be of significance to the field and related fields?

On these grounds, it is possible to expect that this work will prove an effective tool in the field of supramolecular host-guest systems.

How does it compare to the established literature? If the work is not original, please provide relevant references.

From the methodological standpoint, this work relies on well-established concepts pertaining relaxation. There are a number of papers using the same concept, for instance, in proving protein-target interactions, see for instance the work of Fragai et al. on cobalt(II)-containing enzymes (*Angewandte Chemie* 2005) or the work of Hilty et al. on manganese(II) containing enzymes (*J. Phys. Chem. Lett.* 2017). There are a few methodological comments that arise from this (see below). Overall, literature referencing in the first paragraphs seems not completely in focus (and somehow biased towards groups that are less into the center of the paramagnetic NMR development).

Does the work support the conclusions and claims, or is additional evidence needed?

Conclusions are supported by the data.

Are there any flaws in the data analysis, interpretation and conclusions? Do these prohibit publication or require revision?

The data are analyzed properly.

Is the methodology sound? Does the work meet the expected standards in your field?

The metal that the authors considered is a very tricky case. This is because the electron relaxation is relatively slow as compared to other metals, leading to broader NMR lines, and the magnetic susceptibility is small as compared to other metals. I understand from the context that the choice of the metal is dictated by the required reactivity. Yet, if zinc(II) could work structurally as well, other first-row transition metals could work as well, disclosing different data analysis opportunities. In any case, manganese(III) has some magnetic anisotropy, which should also induce shifts on the guest nuclei, besides relaxation, and this could be further interpreted structurally. Substitution with iron(III) could yield larger shifts and slower nuclear relaxation (narrower lines). I would certainly recommend the authors to attempt this substitution.

On the relaxation side, substitution with copper(II), cobalt(II) - assuming it will result in a low-spin state, or vanadium(II) could give a different dynamic range to the measurements. This could be attempted or, at least, discussed.

Is there enough detail provided in the methods for the work to be reproduced?

Yes

Reviewer #2 (Remarks to the Author):

This manuscript by Elemans, Nolte and co-workers reports the use of paramagnetic relaxation enhancement NMR spectroscopy to quantify binding affinities and threading/dethreading parameters for viologen derivatives within a paramagnetic Mn(III) porphyrin host. The authors have previously studied diamagnetic analogues of this system by various characterisation techniques. However, the Mn(III) system was of interest since it is the catalytic system but the paramagnetism of the Mn(III) centre prevented determination of whether the viologen derivatives thread through the host due to the broadness of the NMR signals. This study not only demonstrates that threading takes place but also shows that quantitative data can be obtained to give insight into the threading dynamics of the Mn(III) host compared with the metal-free host.

This is a well-performed and thorough study that will be of interest to the supramolecular community given the number of paramagnetic host systems and challenges associated with their characterisation by NMR spectroscopy. While methods are available for the characterisation of paramagnetic systems with relatively sharp NMR signals, the advantage of the approach in this manuscript is that it can be used even when the signals for the host-guest complex are too broad or missing since it measures the effect of the paramagnetic species on the bulk solvent or free guest. I would recommend publication in Nature Communications with the following minor revisions:

1) While I agree with the statement that the study of paramagnetic systems can be challenging due to the large paramagnetic shifts and short relaxation times, there have been a number of recent reports where paramagnetic inorganic and supramolecular systems have been successfully studied by NMR spectroscopy (examples include: ref. 6, Chem. Eur. J., 2016, 22, 18123-18131; Chem. Commun., 2019, 55, 14426-14429; Inorg. Chem., 2020, 59, 12758-12767; Angew Chem Int. Ed., 2020, 59, 19344-19351; Angew. Chem. Int. Ed., 2021, 60, 22856-22864.). Furthermore, paramagnetic host-guest systems based on cages have been recently studied by Dmochowski using parahyperCEST (e.g. ref. 6 and Inorg. Chem., 2020, 59, 12758-12767) and the Xe exchange dynamics were determined using this method. It would be useful to include references about the state-of-the-art for the characterisation of paramagnetic systems to provide context for this work.

2) It is not clear to me why different polymers and temperatures have been used for comparisons in Tables 1 and 3. For example, in Table 1 an approximate binding constant for H21 and VP is given from two literature references and this binding constant was determined at 23 °C when all other Gibbs free energies were reported at 25 °C.

3) While PRE methods have been used in other fields such as biochemistry, it may not be so familiar to supramolecular chemists. Given that several criteria need to be met (pages 8-9 and SI) for the method to be valid to use and corrections may need to be applied (e.g. for outer sphere effects), it would be useful to provide a how-to-guide for the method so others can apply it to different paramagnetic systems.

Answers to questions of reviewers

We are very pleased with the positive comments of the two reviewers and we thank them for the constructive feedback. The answers to their questions and the changes that have been made are indicated below.

Reviewer 1

From the methodological standpoint, this work relies on well-established concepts pertaining relaxation. There are a number of papers using the same concept, for instance, in proving protein-target interactions, see for instance the work of Fragai et al. on cobalt(II)-containing enzymes (Angewandte Chemie 2005) or the work of Hilty et al. on manganese(II) containing enzymes (J. Phys. Chem. Lett. 2017). There are a few methodological comments that arise from this (see below). Overall, literature referencing in the first paragraphs seems not completely in focus (and somehow biased towards groups that are less into the center of the paramagnetic NMR development).

Answer: We thank the reviewer for pointing this out. The introduction has been rewritten to include more literature on different techniques used to study paramagnetic systems (utilizing hyperfine shifts, hyperpolarization, DOSY, hyper paraCEST NMR, and paramagnetic GEST NMR, see below). The following publications have been added:

Use of hyperfine shifts in paramagnetic systems:

Ott, J. C. et al. Observability of paramagnetic NMR Signals at over 10 000 ppm chemical shifts. Angew. Chem. Int. Ed. 60, 22856–22864 (2021).

Bertini, I. et al. Structural information through NMR hyperfine Shifts in blue copper proteins. J. Am. Chem. Soc. 122, 3701–3707 (2000).

Feng, Y., Roder, H. & Englander, S. W. Redox-dependent structure change and hyperfine nuclear magnetic resonance shifts in cytochrome c. Biochemistry 29, 3494–3504 (1990).

Dissolution dynamic nuclear polarization

Liu, M., Zhang, G., Mahanta, N., Lee, Y. & Hilty, C. Measurement of kinetics and active site distances in metalloenzymes using paramagnetic NMR with ¹³C hyperpolarization. J. Phys. Chem. Lett. 9, 2218–2221 (2018)

Paramagnetic hyperCEST

Du, K., Zemerov, S. D., Hurtado Parra, S., Kikkawa, J. M. & Dmochowski, I. J. Paramagnetic organocobalt capsule revealing xenon host–guest chemistry. Inorg. Chem. 59, 13831–13844 (2020).

Du, K., Zemerov, S. D., Carroll, P. J. & Dmochowski, I. J. Paramagnetic shifts and guest exchange kinetics in ConFe_{4–n} metal–organic capsules. Inorg. Chem. 59, 12758–12767 (2020)

Paramagnetic GEST (host-guest variant of CEST) combining ¹⁹F-MRI and ¹⁹F-NMR

Goren, E., Avram, L. & Bar-Shir, A. Versatile non-luminescent color palette based on guest exchange dynamics in paramagnetic cavitands. *Nat. Commun.* 12, 3072 (2021)

Diffusion ordered spectroscopy (DOSY) NMR

Denis-Quanquin, S., Riobé, F., Delsuc, M.-A., Maury, O. & Giraud, N. Paramagnetic DOSY: an accurate tool for the analysis of the supramolecular interactions between lanthanide complexes and proteins. *Chem. – Eur. J.* 22, 18123–18131 (2016)

P. Crockett, M., Zhang, H., M. Thomas, C. & A. Byers, J. Adding diffusion ordered NMR spectroscopy (DOSY) to the arsenal for characterizing paramagnetic complexes. *Chem. Commun.* 55, 14426–14429 (2019)

Toolbox utilizing standard NMR techniques for characterizing paramagnetic compounds

Lehr, M. et al. A Paramagnetic NMR spectroscopy toolbox for the characterisation of paramagnetic/spin-crossover coordination complexes and metal–organic cages. *Angew. Chem. Int. Ed.* 59, 19344–19351 (2020)

The metal that the authors considered is a very tricky case. This is because the electron relaxation is relatively slow as compared to other metals, leading to broader NMR lines, and the magnetic susceptibility is small as compared to other metals. I understand from the context that the choice of the metal is dictated by the required reactivity. Yet, if zinc(II) could work structurally as well, other first-row transition metals could work as well, disclosing different data analysis opportunities. In any case, manganese(III) has some magnetic anisotropy, which should also induce shifts on the guest nuclei, besides relaxation, and this could be further interpreted structurally. Substitution with iron(III) could yield larger shifts and slower nuclear relaxation (narrower lines). I would certainly recommend the authors to attempt this substitution. On the relaxation side, substitution with copper(II), cobalt(II) - assuming it will result in a low-spin state, or vanadium(II) could give a different dynamic range to the measurements. This could be attempted or, at least, discussed.

Answer: We thank the reviewer for highlighting these points. As part of our current studies, we have chosen to focus on the kinetics of dissociation and the thermodynamic properties of association of manganese porphyrin host-guest complexes in order to get more information on the threading mechanism that we assume to be operative for polymer-substituted viologen guest compounds. Structural information is definitely of high interest, which, for future studies, we aim to further investigate by, e.g., hyperfine shifts or EPR spectroscopy. We are currently performing initial association and dissociation studies for cobalt(II) and iron(III) chloride substituted-porphyrin cages. Using the current temperature range, solvent, and guest (bis(methyl cyclohexyl)viologen), we have

been able to extract association constants for these systems. These preliminary results highlight the applicability of the method to supramolecular systems containing different metal centers. Unfortunately, we have not been able to extract dissociation rate constants using the current conditions. In the case of the iron(III)Cl porphyrin cage and the viologen-guest complex, the electronic relaxation of the metal center is too fast. The cobalt(II) porphyrin cage and the viologen-guest complex are in the fast-exchange regime, and therefore we see that the inner-sphere relaxation rate dominates, preventing the extraction of rate constants in this case. In the future, this could be studied by trying a different solvent (e.g. dichloromethane) or different solvent-mixtures, allowing us to go to lower temperatures or differently substituted viologen-guests containing bulkier side groups (e.g., isononyl or adamantyl groups). For this reason, we feel that discussing these differently substituted metalloporphyrin cages does not fit the scope of the current manuscript. We are planning a more elaborate study involving a range of differently metallated porphyrin cages, including diamagnetic host/guest-complexes such as a zinc(II) porphyrin cage and a nickel(II) porphyrin cage, which we are currently studying using EXSY NMR). To clarify this to the reader we have added a note to the main text, explaining that different metalloporphyrin cages are currently under investigation.

Reviewer 2

While I agree with the statement that the study of paramagnetic systems can be challenging due to the large paramagnetic shifts and short relaxation times, there have been a number of recent reports where paramagnetic inorganic and supramolecular systems have been successfully studied by NMR spectroscopy (examples include: ref. 6, Chem. Eur. J., 2016, 22, 18123-18131; Chem. Commun., 2019, 55, 14426-14429; Inorg. Chem., 2020, 59, 12758-12767; Angew Chem Int. Ed., 2020, 59, 19344-19351; Angew. Chem. Int. Ed., 2021, 60, 22856-22864.). Furthermore, paramagnetic host-guest systems based on cages have been recently studied by Dmochowski using parahyperCEST (e.g. ref. 6 and Inorg. Chem., 2020, 59, 12758-12767) and the Xe exchange dynamics were determined using this method. It would be useful to include references about the state-of-the-art for the characterisation of paramagnetic systems to provide context for this work.

Answer: We agree with this reviewer that a more complete overview of the literature would be useful for the reader. We have added the suggested references encompassing different methods to study paramagnetic species, to the revised version. See also our answer to the first remark of Reviewer 1.

It is not clear to me why different polymers and temperatures have been used for comparisons in Tables 1 and 3. For example, in Table 1 an approximate binding constant for H21 and VP_{var} is given from two literature references and this binding constant was determined at 23 °C when all other Gibbs free energies were reported at 25 °C.

Answer: We thank the reviewer for pointing this out. In our study all binding constants have been determined at 25 °C and the values from earlier literature studies for VP_{var} were given at 23 °C to stay as closely as possible to the literature. However, we have done a recalculation and below we provide tables comparing the energy differences at 23 °C and 25 °C. The conclusion is that there

is an insignificant difference between these two temperatures. In the main text, we have changed the Gibbs values to those calculated for 25 °C and added a note under the table.

Table 1: Gibbs free energies of association for VP_{var} calculated for different temperatures (23 and 25 °C).

Temperature (°C)	Host	Guest	Log(K _a)	Δ G (kcal·mol ⁻¹)
25	H ₂ 1	VP _{var}	7.4	-10.1
23	H ₂ 1	VP _{var}	7.4	-10.0

Table 2: Gibbs free energies of threading for VP_{var} calculated for different temperatures (23 and 25 °C).

Temperature (°C)	Host	#THF-units in VP	Δ G [‡] (kcal·mol ⁻¹)
25	H ₂ 1	34	11.23 ± 5%
23	H ₂ 1	34	11.16 ± 5%
25	H ₂ 1	60	11.61 ± 5%
23	H ₂ 1	60	11.53 ± 5%

While PRE methods have been used in other fields such as biochemistry, it may not be so familiar to supramolecular chemists. Given that several criteria need to be met (pages 8-9 and SI) for the method to be valid to use and corrections may need to be applied (e.g. for outer sphere effects), it would be useful to provide a how-to-guide for the method so others can apply it to different paramagnetic systems.

Answer: We thank the reviewer for suggesting this. A step-by-step how-to guide for using these methods to determine dissociation and association constants has been added to the supporting information (see below) and a sentence in the main text has been added notifying the reader of the availability of the how-to guide.

How-to guide for utilizing PRE relaxometry studies to determine association and dissociation constants in paramagnetic host-guest complexes

Determination of association constants using relaxometry in PRE NMR

To determine the association constant of a host-guest complex of choice, a reporter solvent needs to be chosen. The reporter solvent needs to interact with the host of choice in the form of (weak) binding, and exchange is required to occur. Additionally, the guest of choice should have a stronger binding constant than the reporter solvent. To determine the suitable host concentration in these studies, a test run should be performed first. With an estimate of the association constant in mind, at least three (but more if desired) samples should be prepared. Sample 1 should contain only diamagnetic guest

(determines $R_{1,0}$), sample 2 should contain host and no guest (determines the maximum $R_{1,obs}$), and sample 3 should contain host and the concentration of guest that is estimated to give complete association of the host (determines the minimum $R_{1,obs}$). See below for K_a estimation calculations.

During the test run, the T_1 relaxation time constant of the reporter solvent is measured via the inversion recovery pulse sequence of all samples. The relaxation rate for sample 1 ($R_{1,0}$) is used to calculate $R_{1,p}$ for samples 2 and 3 using Equation G1. Between samples 2 and 3, a sufficient difference in $T_{1,p}$ should be observed (~ 0.3 s minimum or a minimum difference in $R_{1,p}$ of ~ 0.75 , see Figure G1) for the results to be reliable.

$$R_{1,obs} = R_{1,0} + R_{1,p} \quad (G1)$$

Figure G1. Example of a test-run of Mn1/V2 (25 °C, 300 MHz, reporter solvent: chloroform) using 0.3 mM of host and 0 mM of guest for sample 2 ($R_1 = 0.81$ s $^{-1}$) and 13.5 mM of V2 for sample 3 ($R_1 = 0.53$ s $^{-1}$).

When the difference in $T_{1,p}$ between samples 2 and 3 is insufficient, a couple of parameters may improve the results of these studies: a higher concentration of host, varying the temperature, or measuring at a lower magnetic field.

After a successful test run, the samples can be prepared separately, varying in guest concentration, or the guest can be added in between measurements. The latter method is preferred if a limited amount

of compound is available but is not desirable if it is known that the guest requires a longer time to form the host-guest complex.

After measuring the T_1 -values of the sample(s), the $R_{1,p}$ ($1/T_{1,p}$) is plotted against the added concentration of guest to give a typical binding curve (Figure G2).

Figure G2. Example of a titration curve of Mn1/V2 (25 °C, 300 MHz, reporter solvent: chloroform) using 0.3 mM of host and up to 13.5 mM of V2.

Table G1. Plotted values of concentrations guest (V2) and $R_{1,p}$ values used in Figure G2.

Conc. guest (M)	$R_{1,p}$ (s^{-1})
0	0.806634
0.00005	0.758904
0.0003	0.634244
0.0006	0.58252
0.00105	0.562054
0.001725	0.553691
0.004875	0.549571
0.0081	0.54017

0.0135

0.527958

The obtained R_1 values can be used to calculate the K_a , using OriginLabs (or any similar software), using the following custom fitting equations:

Function type in OriginPro 2022: LabTalk Equations

Independent variables: Gt

Dependent variables: Robs

Parameters: Ka, Rb

Constants: Ht, Rf

Function body (Dependent variables: Robs):

a = Ka;

b = 1 - Ka*Gt + Ka*Ht;

c = -Gt;

G = (-b + sqrt(b^2 - 4*a*c))/(2*a);

Robs = ((Rb - Rf)*(Gt - G))/Ht + Rf;

Constants (for this specific fit):

Ht = 0.0003

Rf = 0.806634

Figure G3. Example of a titration curve of Mn1/V2 (25 °C, 300 MHz, reporter solvent: chloroform) using 0.3 mM of host and up to 13.5 mM of V2 after fitting.

In the equations K_a = the association constant (M^{-1}), $G_t = [G]_{total}$ = the total guest concentration (M), $H_t = [H]_{total}$ = the total host concentration (M), $R_b = R_{bound}$ = the relaxation rate of the bound species (s^{-1}), $R_f = R_{free}$ = the relaxation rate of the unbound species (s^{-1}). For further information and earlier studies on enzymatic systems, see Bertini et al.¹.

K_a calculations

To calculate the expected percentage of bound host, the desired host concentration ($[H]_{tot}$, M), the ligand concentration ($[G]_{tot}$, M), and the association constant estimation (K_a , M^{-1}) are used. The concentration of bound host ($[HG]$, M) is calculated by the quadratic equation G2.

$$[HG] = \frac{-b - \sqrt{b^2 - 4ac}}{2a} \quad (G2)$$

in which

$$a = 1, b = -\left([G]_{tot} + [H]_{tot} + \frac{1}{K_a}\right), \text{ and } c = [H]_{tot} \cdot [G]_{tot}$$

It is converted to the percentage of bound host with equation G3:

$$\frac{[\text{HG}]}{[\text{H}]_{\text{tot}}} \cdot 100 = \% \text{bound host} \quad (\text{G3})$$

Determination of dissociation rate constants using relaxometry in PRE NMR

To determine the dissociation constants using this method, a set of requirements needs to be met before reliable results can be obtained.

A sample should be prepared containing the paramagnetic host and the diamagnetic guest (sample 1). Subsequently, the T_1 relaxation time constant at a given temperature is measured by the inversion recovery pulse sequence and the T_2 by either the CPMG or PROJECT-CPMG pulse sequence. Since these observed R_1 and R_2 values are the sums of the paramagnetic and the diamagnetic contributions (equation G4), another sample should be prepared containing only the diamagnetic guest (sample 2) of which the R_1 and R_2 values are measured (R_0). From the obtained values of samples 1 and 2, the paramagnetic contribution to the relaxation rates (R_p) can be calculated.

$$R_{1_{\text{obs}}} = R_{1,0} + R_{1,p} \quad (\text{G4})$$

The obtained values for $R_{1,p}$ and $R_{2,p}$ are then used to determine whether the following requirements are met:

1. R_p should increase at higher temperatures: $\frac{\partial R_p}{\partial T} > 0$
2. R_p is independent of the field strength: $\frac{\partial R_p}{\partial \omega} = 0$
3. $R_{1,p}$ should be approximately equal to $R_{2,p}$: $\frac{R_{2,p}}{R_{1,p}} \approx 1$

Figure G4. Results obtained if the three listed requirements are met. **a.** Indication of the different dominating factors governing exchange at different temperatures (low T is left, high T is right). At low temperatures where there is no exchange, the outer-sphere interactions are dominating; at higher temperatures the slow and intermediate exchange regime holds, and the exchange is dominating; and in the fast exchange regime the inner-sphere interactions dominate. **b.** Graph of $\ln(R_{1,p})$ against the temperature, displaying a positive slope with increasing temperature. **c.** $R_{1,p}$ values plotted against temperature comparing 300 MHz and 500 MHz experiments. **d.** Ratio of $R_{2,p}$ and $R_{1,p}$ plotted against temperature.

Testing the validity of each requirement

Requirement 1: $\frac{\partial R_p}{\partial T} > 0$

a) Choose two different temperatures and measure the T_1 and T_2 time constants at each temperature point for samples 1 and 2. Ideally, the difference should be >20-30 K.

b) Calculate and plot R_p versus temperature. A positive slope indicates that exchange occurs in an appropriate regime for extracting rate constants. A negative slope could indicate a fast-exchange regime (inner-sphere contributions dominate) or a very slow-exchange regime (outer-sphere contributions dominate) but is overall unsuitable for extracting rate constants. See Mildvan et al.² for an elaboration on the theory.

Requirement 2: $\frac{\partial R_p}{\partial \omega} = 0$

a) R_p should be determined using NMR spectrometers with different magnetic field strengths. A 300 MHz and a 500 MHz spectrometer were used for the present studies.

b) Calculate and plot R_p versus field strength (ω). If a non-zero slope is observed, it is evident that nuclear relaxation rather than chemical exchange is dominating R_p , and extraction of rate constants becomes challenging.

Requirement 3: $\frac{R_{2,p}}{R_{1,p}} \approx 1$

a) Use the data acquired in *Requirements 1* and *2* to calculate $R_{2,p}/R_{1,p}$. If the ratio is not ~ 1 then either inner-sphere or outer-sphere interactions are dominant.

If the ratio between $R_{2,p}$ and $R_{1,p}$ starts to increasingly deviate from 1 upon lowering the temperature, an extra sample can be prepared to account for the outer-sphere interactions (sample 3). These interactions are non-bonding interactions between the free diamagnetic guest and the paramagnetic host and arise solely from the presence of a paramagnetic species. Therefore, a new host should be synthesized which resembles the original host as much as possible, but in which the binding pocket is blocked or in which exchange, and therefore association, is prevented. After correcting $R_{1,p}$ and $R_{2,p}$ for the outer-sphere contribution, *Requirement 3* should now be met.

Extracting Rate Constants Through Mathematical Simplification

In the case that all requirements are met, equation G5 can be simplified to equation G6. This is possible as the above requirements indicate that the residence time of the guest near or on the metal-center (τ_M, s) is much greater than T_M (inner-sphere relaxation constant) and outer-sphere effects (R_{OS}). The observed dissociation rate constant (k_{obs}) can be calculated from the inverse of τ_M .

$$R_p = \frac{f}{\tau_M + T_M} + R_{OS} \quad (G5)$$

$$R_p = \frac{f}{\tau_M} + R_{OS} \quad ; \quad \tau_M^{-1} = k_{obs} \quad (G6)$$

in which “ f ” is the mole-fraction of the ligand bound to the host. Under saturation conditions this is simply the total host to total guest ratio. However, the fraction of the host bound by guest must be determined for any given ligand/host sample, using the determined association constant (K_a) for the system.

When *Requirement 3* is met under all measured temperatures and sample 3 is not required, then the R_{os} term can be ignored and the equation is further simplified to equation G7.

$$R_p = \frac{f}{\tau_M} \quad (G7)$$

When there is no ligand dependence on the exchange process, k_{obs} is the true dissociation rate constant k_d . However, when a ligand dependence is observed, k_{obs} must be corrected for the order of the dependence. Therefore, the ligand dependence should be tested for studied systems by determining the τ_M values at different ligand concentrations.

From the obtained k_d at various temperature points, an Eyring plot can be constructed, and the dissociation parameters ΔH^\ddagger , ΔS^\ddagger , ΔG^\ddagger can be determined.

Fast Exchange Regime

If the system is in the fast-exchange regime, the inner-sphere relaxation dominates over the exchange, and it is not viable to extract rate constants under these conditions. However, when the behaviour is observed to be in the intermediate exchange regime, where it transitions from slow to fast (Figure G4), the rate constant can still be extracted provided sufficient fast-exchange temperature points can be acquired. The value of R_M can be calculated using equation G8.

$$R_{obs} = \chi_{HG} \cdot R_M + \chi_G \cdot R_0 \quad ; \quad R_M = T_M^{-1} \quad (G8)$$

in which χ_{HG} and χ_G are the mol fractions of bound and free guest, respectively. After measuring R_M at several temperatures, $\ln(R_M)$ is plotted vs temperature and R_M (and thus T_M) is extrapolated for temperatures where R_M is not dominating (in which $\frac{\partial R_p}{\partial T} > 0$). The calculated values for T_M can then be used to calculate τ_M using equation G5.

Figure G5. Schematic depiction of the transition between the intermediate-exchange rate (positive slope) and the fast-exchange rate (negative slope). The $R_{1,p}$ (s^{-1}) is plotted against the temperature (K). The temperature range which yields a negative slope for the change in $R_{1,p}$ is indicated in grey. This range can be used to calculate R_M .

References

1. Bertini, I., Fragai, M., Luchinat, C. & Talluri, E. water-based ligand screening for paramagnetic metalloproteins. *Angew. Chem. Int. Ed.* **47**, 4533–4537 (2008).
2. Mildvan, A. S. & Cohn, M. Aspects of enzyme mechanisms studied by nuclear spin relaxation induced by paramagnetic probes. in *Advances in Enzymology and Related Areas of Molecular Biology* 1–70 (John Wiley & Sons, Ltd, 1970). doi:10.1002/9780470122785.ch1.

REVIEWER COMMENTS

Reviewer #2 (Remarks to the Author):

The authors have satisfactorily addressed the reviewers' comments. The context of the work has been improved through the inclusion of additional references and the how-to guide will facilitate the implementation of the method by others. I agree with reviewer 1 that it would be good to discuss/include the study of systems based on other paramagnetic metals and the authors provide a justification why this falls outside the scope of this study based on their undergoing investigations.